# Big Data and Machine Learning to Improve European Grapevine Moth (*Lobesia botrana*) Predictions

**DOI:** 10.3390/plants12030633

**Published:** 2023-02-01

**Authors:** Joaquín Balduque-Gil, Francisco J. Lacueva-Pérez, Gorka Labata-Lezaun, Rafael del-Hoyo-Alonso, Sergio Ilarri, Eva Sánchez-Hernández, Pablo Martín-Ramos, Juan J. Barriuso-Vargas

**Affiliations:** 1Department of Agricultural Sciences and Natural Environment, AgriFood Institute of Aragon (IA2), University of Zaragoza, Avenida Miguel Servet 177, 50013 Zaragoza, Spain; 2Department of Big Data and Cognitive Systems, Instituto Tecnológico de Aragón, ITAINNOVA, María de Luna 7-8, 50018 Zaragoza, Spain; 3Departamento de Informática e Ingeniería de Sistemas, Instituto de Investigación en Ingeniería de Aragón (I3A), Universidad de Zaragoza, María de Luna 1, 50018 Zaragoza, Spain; 4Department of Agricultural and Forestry Engineering, ETSIIAA, University of Valladolid, Avenida de Madrid 44, 34004 Palencia, Spain

**Keywords:** *Lobesia botrana*, pest monitoring, predictive models, IoT, weather data, data-driven models, machine learning, integrated pest management

## Abstract

Machine Learning (ML) techniques can be used to convert Big Data into valuable information for agri-environmental applications, such as predictive pest modeling. *Lobesia botrana* (Denis & Schiffermüller) 1775 (Lepidoptera: Tortricidae) is one of the main pests of grapevine, causing high productivity losses in some vineyards worldwide. This work focuses on the optimization of the Touzeau model, a classical correlation model between temperature and *L. botrana* development using data-driven models. Data collected from field observations were combined with 30 GB of registered weather data updated every 30 min to train the ML models and make predictions on this pest’s flights, as well as to assess the accuracy of both Touzeau and ML models. The results obtained highlight a much higher F1 score of the ML models in comparison with the Touzeau model. The best-performing model was an artificial neural network of four layers, which considered several variables together and not only the temperature, taking advantage of the ability of ML models to find relationships in nonlinear systems. Despite the room for improvement of artificial intelligence-based models, the process and results presented herein highlight the benefits of ML applied to agricultural pest management strategies.

## 1. Introduction

Agriculture faces the challenge of increasing its productivity while reducing its environmental impact [1,2]. Nonetheless, agricultural systems are affected by different problems, such as environmental conditions, soil characteristics, water availability, and pest infestation. The latter has traditionally been managed through the knowledge and experience of the farmers themselves [3,4].

Today, agriculture is undergoing a major transformation in the collection and use of large amounts of data to make smarter decisions [5]. The applicability of new trends such as the Internet of Things (IoT) and Information and Communication Technologies (ICT) to agricultural activities enables farmers to adopt an approach based on the analysis of data from their farms to improve decision-making efficiency and mitigate pest risks [3,6]. Recent developments regarding open access to data, coupled with the unprecedented growth in the volume of data, referred to as Big Data (BD), have led to a shift in focus toward methods aimed at effectively managing such data for use in agri-environmental research [7]. In this sense, both traditional statistical techniques and Machine Learning (ML) techniques can be used to convert collected BD into timely and valuable information [8].

The grapevine moth, *Lobesia botrana* (Denis & Schiffermüller) 1975 (Lepidoptera: Tortricidae) is one of the main pests of grapevines [9,10,11,12], responsible for high productivity losses in some vineyards worldwide. Figure 1 shows how this pest can be identified in the field at different stages of development. *L. botrana* presents a facultative diapause [13,14,15,16], which results in a variable number of flights and generations per year, depending on the temperature and photoperiod [15,17]. *L. botrana* is multivoltine; in Mediterranean latitudes, it has three generations [15].

*L. botrana* causes different types of damage (Figure 2). On the one hand, direct damage caused by the pest occurs during the first generation, destroying a certain number of inflorescences, which generally does not affect crop yield [18]. In the second and third generations, the larvae directly attack the berries, causing more significant harvest losses [19]. Moreover, regarding indirect damage, in the second generation, the attack on berries favors the presence of *Botrytis cinerea* Pers.: Fr (*Botryotinia fuckeliana* (de Bary) Whetz.) [10,13,16].

In terms of crop protection, it has been experimentally demonstrated that integrated pest management (IPM) [20] is more effective than classical methods based solely on biological or chemical control alone [21]. IPM is defined as a long-term control strategy that combines biological, cultural, and chemical methods to reduce pathogenic populations to tolerable levels so that pests do not reach an economic threshold of damage [22]. In this line, as occurs with other pests, *L. botrana* control strategies are based on monitoring populations and applying control measures (such as the sex pheromone-based mating disruption technique, mentioned later in this section, or phytosanitary treatments, when necessary) to avoid reaching the aforementioned level of economic damage [23].

It is worth pointing out that this type of monitoring-based population tracking has sensitive aspects, such as the significant amount of time needed to carry it out, as well as a high level of technical expertise [4]. In addition to this, an insufficiently high monitoring frequency may represent a significant loss of information between field inspections [24]. In this sense, the potential of predictive models of future pest behavior makes modeling a powerful resource to complement monitoring, as well as being particularly useful for decision support systems (DSS), which help farmers to make decisions on crop protection measures [25,26]. According to Nansen et al. [27], a large part of the production systems are expected to face an increase in pest pressure, and therefore in potential phytosanitary applications. Because of this, the assessment of the sustainability of viticulture under future conditions, both from an environmental and economic point of view, can be considered of utmost importance for winegrowing operations [28]. Consequently, the importance of the use of DSS lies in their viability to increase the sustainability of agricultural production, which can lead to a reduction of the effects of agrochemicals on the environment and their economic cost at the farm level [29,30].

Considering that agricultural crops are managed biological systems, the utilization of applied mathematics developed for biological systems can be of great use in pest control through modeling and simulation, as Plant et al. [31] explained at the time. Nowadays, the role that modeling can play in plant health continues to arouse interest among entomologists and environmental scientists [32]. Bearing in mind that the objective of an optimal application of phytosanitary products is not only to control a pest but to predict and mitigate its appearance [2], predictive models are considered essential for efficient crop protection methods [33,34,35], because of their usefulness in deciding if and when treatments are necessary.

Since insects are ectothermic organisms, it has been shown that there is a direct relationship between their life cycle and abiotic factors, such as climatic factors (viz. temperature, relative humidity, wind direction, rainfall, photoperiod, and insolation) [24]. In the case of *L. botrana*, temperature is the variable with the greatest influence on its development [9,12,13,14,15,17,32,36,37,38,39,40,41]. Hence, the Touzeau predictive model for the development of *L. botrana* [42], developed in the Toulouse region (France), is based on the calculation of thermal integrals through temperature accumulation above a set threshold value. Flight monitoring of this pest, which is carried out via pheromone trapping of adults, provides potentially useful data to verify Touzeau’s predictions [38].

In recent years, the sex-pheromone-based mating disruption technique has been successfully implemented for *L. botrana* control [43,44], with a significant reduction in the use of chemical insecticides and higher protection of non-target insects [45,46]. This method is based on the use of pheromones emitted by insects, which they use as chemical messengers. The best known and most widely used are the sex pheromones emitted, in this case, by the females of Lepidoptera to attract males and facilitate mating. Thus, the strategy is to saturate the air with the female’s sex pheromone by arranging diffusers, disrupting communication between the female and the male [47]. As a consequence, mating is hindered or delayed, leading to a drastic and gradual reduction in the oviposition of fertile eggs in later generations, causing a noticeable decline in the pest population [46,48]. This technique began to be implemented in 2012 in the province of Zaragoza (Aragón, Spain), the study area on which this work focuses. At the present time, in Aragon, *L. botrana* is controlled almost exclusively using mating disruption (although, if necessary, phytosanitary treatments are carried out on an ad hoc basis).

The research presented herein focuses on the analysis of a correlation model between weather conditions and *L. botrana* development, analogous to the aforementioned Touzeau predictive model. This analysis is centered on three protected designations of origin (PDO), viz. Cariñena, Campo de Borja, and Calatayud, located in the province of Zaragoza. Given that the fit of the Touzeau model is not sufficiently accurate in this area, its improvement would entail two main advantages: (i) A contribution to the preventive control of *L. botrana* in case the established sex pheromone-based mating disruption system becomes ineffective (at present, the mating disruption method is still effective, but in other pests, it has been observed that males learn to differentiate between the pheromone of the diffusers and that of the females, which is much more complex); and (ii) the possibility to tackle the adaptation of this pest to the modification of meteorological conditions in the area due to climate change, which has already begun to be observed by several authors [9,15,16,40,49,50]. From both points of view, the optimization of the model would be very useful to schedule an efficient treatment calendar. It is therefore proposed to improve the current predictive model through the use of Artificial Intelligence (AI)-based systems, particularly using ML models based on neural networks, whose highly hierarchical structure and great learning capacity allow for particularly good classifications and predictions [1].

## 2. Results

A total of 10,000 models were trained. For all of them, the projection of the data for the last two weeks was considered input data. During training, these models are able to capture knowledge and are expected to find non-linear relationships with more than one variable. The variables identified as most relevant by each model were extracted from their descriptions. The best-performing trained model was an Artificial Neural Network (ANN) of four layers whose composition is shown in Table 1.

Regarding the aforementioned variables, the best-performing model considered the following variables as most relevant:The Touzeau index.The Chilling index.The rainfall (in this case, calculated using daily data or accumulating the half-hourly samples, considering the starting date of dormancy as the start date to accumulation).The longitude.

The results of the F1 metric for the classic Touzeau models and the ML models were obtained after comparing the day of the year (DOY) when the flight peaks were observed in the field with the DOY predicted by the models. Table 2 shows the comparison of observed flight peak DOY with the DOY predicted by the Touzeau and ML models for the 2008−2011 seasons. These results highlighted, firstly, that the Touzeau model did not provide a good fit in the study area, yielding a 0.03 F1 score, which is very low. On the other hand, the ML models resulted in much higher F1 scores, of up to 0.63.

## 3. Discussion

### 3.1. On the Model Performance

The F1 score reaches its best value at 1, while 0 is the worst score [51]. It should be clarified that there is no specific threshold value for considering an F1 score as ‘good’, so the model that produces the highest F1 score is generally considered the best. In this work, the Touzeau model obtained a 0.03 F1 score, which is very low compared with that obtained by the ANNs (of up to 0.63), so ML models can be regarded as more accurate.

This F1 score value of 0.63 may be assessed in a similar way as Sepúlveda et al. [52] did in their research: They concluded that the F1 score of 0.6 that they obtained could be interpreted as a low false negative rate and a similar false positive rate, with good compensation between errors, confirming a good F1 score metric.

Another interpretation would be based on the approach proposed by Nieto et al. [53], who used F1 to compare the results of two classification models developed for monitoring phenology. The best model under the F1 metric was 0.52 times better than the worst one. Although the scenario is not the same as the one for our study (given that their models were similarly developed and here we are comparing a classical model with an elaborated ANN), the same reasoning would suggest that the obtained ML model was 20 times better than the Touzeau model.

### 3.2. Advantages of ML Models and Most Relevant Weather Variables Identified by the Best-Performing ANN

As noted above, the Touzeau model takes into account the daily temperature record as the only parameter, i.e., it does not take into account the rainfall, relative humidity, or any other parameter collected by weather stations. In this sense, the Touzeau model considers all conditions optimal for the development of *L. botrana*. However, according to Stellwaag [54], the highest population levels of this pest, and therefore the most serious economic damage, are systematically recorded in areas with conditions considered optimal for this insect (an average annual temperature of around 20 °C and a relative humidity of around 70%). From this perspective, it can be explained that at times of unsuitable climatic conditions (low humidity, extreme temperatures), the species suffers a slowdown in its physiological processes and even adult mortality. The ML models, on the other hand, consider all the parameters recorded by the weather stations (Appendix A). In addition, they can consider the 48 daily inputs corresponding to the data recorded every 30 min. In this way, ML models calculate the contribution of each of the 48 parts of the day to determine the daily contribution to cumulative indices and values. Furthermore, BD technologies allow for improving the quality of information used to train AI models, balancing skewed datasets (e.g., in our case, the number of samples with flights is lower than the number not having), thus reducing the risk of overfitting [55]. In addition, once built, the way the ML models are trained allows them to be easily updated with new data to adapt them to potential new scenarios.

Following on from the ability of ML models to consider all the weather parameters discussed above, the best-performing ANN considered temperature (Touzeau and Chilling indexes), rainfall, and longitude as the most relevant variables. As mentioned above, many authors have addressed the influence of temperature on the development of *L. botrana*. Others found that both temperature and rainfall were the variables that most influenced this pest distribution [12]. In this same line, Comșa et al. [50] found that warmer and drier seasons were more favorable. Zhan et al. [56] also explained that high temperature and low humidity provide optimal conditions for *L. botrana*, while rain together with low-temperature conditions seems to reduce the mating frequency and, subsequently, egg production.

In addition, previous works considered the relationship between low temperatures, on which the chilling index is based [57], and *L. botrana*. Andreadis et al. [14] studied the cold hardiness of this pest. Other research works pointed out that very low temperature, experienced by larval instars, tends to avert pupal diapause, with a very prolonged development of larvae with low temperatures [17]. Zhan et al. [56], on their side, highlighted that low temperatures had a great influence on the occurrence of this pest.

Regarding longitude, some authors analyzed the correlation between this parameter and other Lepidoptera. Khaghaninia et al. [58] performed their study with *Cydia pomonella* (Linnaeus) 1758 (Lepidoptera: Tortricidae), belonging to the same order and family as *L. botrana*, and their results showed a low correlation. On the other hand, Zhou et al. [59] found a strong relationship between this variable and another insect of this same order, viz. *Parocneria orienta* (Chao) 1978 (Lepidoptera: Erebidae); in particular, these authors noted that the hazard centroid shifted significantly with respect to longitude.

### 3.3. Applicability of the Developed ML Models

The developed ML model can help winegrowers of the three PDOs make better decisions on crop protection measures against *L. botrana*, relying on more accurate model predictions to carry out treatments only if and when they are needed. In this regard, once the model predicts a risk in terms of a pest infestation, and in terms of the field implementation of measures based on that prediction, the cost of taking or not taking measures should be assessed in relation to the risk raised.

Although, at present, *L. botrana* population density remains below an economic threshold of damage in those areas where the mating disruption method using sex pheromones has been implemented (which, as noted above, is still effective), the optimization of the predictions could allow winegrowers to improve the preventive control of *L. botrana* in case the mating disruption method becomes ineffective. Furthermore, it should help farmers to deal with the adaptation of this pest to the modification of meteorological conditions in the area due to climate change.

### 3.4. Room for Improvement of the AI-Based Models and Future Work

Based on results from previous work [60,61], an ANN was selected for training, but other more complex models could be trained to see if they would be able to obtain better results.

Although many models in grapevine entomology refer to a single species, the future of modeling in pest management should move towards global frameworks, covering more aspects of the problem (e.g., economics, crop yields, etc.), where the insect pest becomes part of the whole. For example, it may be necessary to choose the right timing for insecticide spraying against two different pests that overlap in the same season, and if both pests are predicted by a model, it would be easier to choose the best active substance, which would make the decision more environmentally friendly and cheaper [62].

Despite the outlined room for improvement of AI-based models, the process and results presented in this work highlight the benefits of ML applied to plant health strategies and its potential contribution to increasing the sustainability of agricultural activity, through a lower environmental impact as well as a lower economic cost of production (e.g., as a result of determining the right time to initiate insecticide application to control a pest or, preferably, using IPM tactics such as sexual confusion).

As part of the GRAPEVINE project (hiGh peRformAnce comPuting sErvices for preVentIon and coNtrol of pEsts in fruit crops), in which the present study is framed, further research is planned primarily along two lines of work: One will be validating how the ML models perform with data collected after the implementation of the mating disruption method, i.e., from 2012 to 2022, after the mating disruption method implementation. The second main issue to tackle in the future is related to testing the performance of the models in other geographical areas beyond the study area considered in this work.

## 4. Materials and Methods

This section introduces the field sites and the study period. How the *L. botrana* flight monitoring was performed is also presented, as well as the weather data used. The section is completed with a description of Touzeau and data-driven models.

### 4.1. Monitoring Field Sites and Study Period

The vineyards used as control fields are located in three PDOs (Cariñena, Campo de Borja, and Calatayud) in the province of Zaragoza (Aragón, Northeastern Spain). The study area, shown in Figure 3, is characterized by a continental climate, with the typical wind of the region—called ‘*cierzo*’—as the most outstanding characteristic, being a cold and dry north-western wind, with a Mediterranean summer influence. The monitoring sites are commercially cultivated vineyards belonging to wineries and cooperatives of the PDOs. The activity of these producers in terms of plant health is coordinated through the RedFAra network [63]. This network has a database repository with historical records of in situ monitoring and pest incidence. Thus, *L. botrana* flight data from the monitoring sites were available for the study period.

In the study area, the sex-pheromone-based mating disruption technique began to be implemented for *L. botrana* in 2012. Therefore, the study period selected for this work extends from 2004 to 2011 to avoid confusing and biased information on pest behavior. A total of 172 monitoring sites were used for this work during the entire study period (2004−2011).

Figure 4 (left) shows the location of the monitoring sites. Data from the cadaster of the Spanish registry [65] and the Aragon Open Data portal [66] provided the geographical description of the monitoring points, also allowing, through their location, to link each point to the nearest weather station as well as to information related to their geopositioning [60].

### 4.2. Lobesia botrana Flight Monitoring

Data from *L. botrana* flight activity at the monitoring sites used in the study belong to the repository of the RedFAra database [63]. Delta traps were placed one meter above the ground, one per hectare, with a gummed bottom along with a synthetic sex pheromone (E/Z-7,9-dodecadienil acetate) to monitor the flight activity of *L. botrana* males during the study period (Figure 5). Traps were placed during the whole vegetative period of the crop, and captures were checked every week. Pheromones were renewed every 6 weeks, according to the effective duration indicated by the supplier (OpenNatur SL, Lleida, Spain). Data on the captures over time provided information on the time of occurrence and duration of each generation at the monitoring sites.

The *L. botrana* life cycle begins when the first emerging adults from pupae that remained in diapause during the winter are detected. When sustained captures are encountered in pheromone delta traps, the first flight is considered to have taken place. This flight is estimated to end when the number of captures trends to zero. Further capture records should follow the same behavior, although the timing of these records will vary with the increase in average temperatures as spring and summer advance [67]. Data from these captures indicate the maximum flight moments of each generation, called ‘flight peaks’. The information provided by flight monitoring was analyzed together with the meteorological data records collected by weather stations.

### 4.3. Weather Data

The Agroclimatic Information System for Irrigation (SiAR) network, operated by the Spanish Ministry of Agriculture, has weather stations throughout Spain, with 49 of these stations located in Aragon [68]. Weather data stored in the SiAR database were used in this work to analyze the modeling of *L. botrana*. These data correspond to the SiAR stations listed in Appendix A, whose locations are shown in Figure 4 (right). Modeling was performed by managing, between raw and processed data, 30 GB of data updated every 30 min.

The weather data from the stations were combined with data from field observations about flight monitoring. This was performed, on the one hand, to train the ML models—and then to make the predictions—and, on the other hand, to assess the accuracy of both the Touzeau and the ML models. The selection of the most representative weather station for a given vineyard was based on the following rules:First, valid stations were selected for each year: A station was considered valid when, for the year in question, it provided data for at least 90% of the days of the year and the number of days that had less than 90% of the total samples per day (i.e., 48 when the sampling frequency was 30 min) was less than 90%.Once the set of stations was defined for a year, the field observations from a vineyard in that year were combined with the climate data from the nearest station.

The weather stations provided data during the study period on the temperature, relative humidity, precipitation, wind speed and direction, and radiation, with daily and semi-hourly (every 30 min) records (Appendix A).

When a data gap was detected (e.g., station Z22 did not provide data from 18:30 to 20:30 on 25th September 2021), the missing data were replaced. This was achieved using data provided by the European Centre for Medium-Range Weather Forecasts (ECMWF) ERA5 [69], accessed via the Open-Meteo Historical API [70]. ERA5 is the fifth-generation ECMWF atmospheric reanalysis of the global climate and is produced by the Copernicus Climate Change Service (C3S) at ECMWF. It provides hourly data readings on a 30 km grid, which had to be transformed to fit the semi-hourly (every 30 min) SiAR records. For that, all variables were copied into two records, 00 min and 30 min, except for the accumulated precipitation, with which a single register was divided by 2.

Regarding the recorded temperatures, and to analyze the correlation between these and the development of *L. botrana*, the daily maximum (T_M_) and minimum (T_m_) were adjusted to effective temperatures [71]. This was due to the fact that, when working with cumulative indexes, negative temperatures could alter the real value of these indexes. Therefore, to avoid this potential disruption, any temperature below 0 °C was adjusted to the effective temperature of 0 °C.

### 4.4. Touzeau Model

Classical models of *L. botrana*, such as the Touzeau model, are based on temperature accumulation (in °C), which is considered to be the most influential environmental factor on the seasonal behavior of this pest. Degree-day accumulation methods using thermal integrals are the most commonly used methods to analyze the relationship between insect development and temperature [72]. These methods are based on the amount of heat energy accumulated daily above a minimum temperature threshold called the base temperature (T_B_). The Touzeau model uses the Touzeau index to calculate the accumulated thermal integral for *L. botrana*, based on the daily maximum (T_M_) and minimum (T_m_) temperatures, and sets T_B_ = 10 °C [42]. In the present work, in addition to T_B_, a temperature limit of 30 °C (above which the temperature accumulation stops) was taken into account, in agreement with Gabel et al. [73], Del Tío et al. [74], and Gallardo et al. [75]. Therefore, all temperatures with values above 30 °C used to calculate the Touzeau index were adjusted to 30 °C.

The Touzeau model index function is mathematically expressed by:(1)$$Tou=\sum_{\mathrm{DOY}=1}^{n}\left(\frac{T_{M}+T_{m}}{2}-T_{B}\right),$$
where DOY refers to the day of the year or day of the Julian calendar. Thus, the temperature accumulation for the Touzeau index starts on January 1st (DOY = 1).

Appendix A shows the different developmental stages of *L. botrana* for the three generations, together with the cumulative daily temperatures needed to reach them, expressed according to the Touzeau model index (*Tou*). Each of the stages has its calorific requirements. The thermal integral accumulated from a starting point using the *Tou* is used to predict when a stage will be reached. The monitoring methods followed in this paper focus on flights of adult generations. As shown in Appendix A, the accumulated *Tou* for *L. botrana* generations were 125 for 1st G, 500 for 2nd G, and 950 for 3rd G. In this way, to assess the accuracy of the Touzeau model, the DOY when the maximum population of each generation takes place in the field was identified according to flight peaks. Then, this DOY was compared with the DOY when the accumulated *Tou* reached the values indicated by the model for each generation.

### 4.5. Data-Driven Models Used in This Study

Data-driven models are capable of extracting valuable insights from huge amounts of data without the need to define a law or equation to explain a given phenomenon [76]. To do so, they apply BD, analytics technologies and AI algorithms for the automation of data capturing and processing. They can combine heterogeneous multi-sourced data for discovering, understanding, and evaluating new models, which represent relationships that are too complex to be found by mathematical ones [77]. In our context, these computational technologies make it possible to use climatic variables, such as the maximum temperature, minimum temperature, humidity, rainfall, or wind speed, which are known factors influencing pests, but whose influence is not yet well understood [55].

Fenu et al. [55] and Domingues et al. [77] performed a review of the AI, ML, and Deep Learning (DL) algorithms used by different authors for creating pest models. The most popular models are those that use only climate data [55]. Several algorithms with different computational complexity have been used, from Logistic Regression to Convolutional Neuronal Networks (CNN), passing through Support Vector Machines (SVM), Support Vector Regression (SVR), Multi-Linear Regression (MLR), Random Forest (RF), and ANN.

Usually, the predicted variable is a binary variable representing the presence or absence of the pest. The performance of the models is measured using accuracy, precision, recall, or F1 metrics. However, due to the great heterogeneity of experimental conditions (i.e., approaches, datasets, parameters, and performance metrics), it is difficult—if not unreliable—to make a systematic comparison of the performance of approaches presented in different papers [55].

For our study, and based on the results presented in previous work [60], we created a model using an ANN. The aim of the model obtained was to predict the probability that a given date, within a 2-week horizon, is the day of flight peak for each of the generations of *L. botrana* in a given season.

To train (and use) the models, the following transformations were performed on the SiAR and RedFAra datasets:For each monitoring site and season pair, the day of the flight peak for each generation was labeled. This was the variable to be predicted.Growing degree days [78], Touzeau [42], and Chilling [57] indexes were calculated based on both the proportional accumulation of half-hourly samples and daily observations.The aforementioned indexes were calculated considering different accumulation start dates, viz. January 1st, February 1st, or the starting date of dormancy, calculated as the first day of Autumn when the temperature remains under 10 °C. This date was considered to try to increase the accuracy of grapevine phenology predictions [79].Registered radiation values.Wind direction and speed were transformed into daily indexes, calculated using the following rules: wind direction was classified into one of eight categories (N, NE, E, SE, S, SW, W, or NW), and the average speed per day was calculated for each of those categories.The raw weather data were also provided to build the models.The weather data for the 14 days before a given date were projected horizontally so that the model could make predictions over a 2-week horizon.

The models were trained using Scikit-learn [80], Keras [81], and Optuna [82] on Python [83]. As noted above, based on results from previous work [60], an ANN was chosen for training. The selection of the data for testing and validation was performed considering data from different monitoring sites over different years during the study period. The ANN training process used the hyperparameters shown in Table 3.

### 4.6. Model Performance Assessment

The F1 score performance metric was used to assess the accuracy of both the Touzeau and ML models. Although it is slightly more difficult to calculate for the Touzeau model results than other metrics such as R^2^, it has been used by several authors to compare the results of classification models [53,84,85]. F1 is a widely-used metric for assessing the performance of binary classification problems and for comparing the results obtained by different models. It is a good metric when the performance in terms of true positive results is evaluated. However, F1 does not depend on true negatives and is sensitive to the distribution of the labels [86]. In addition, it does enable the user to consider the cost of failing in the classification.

The F1 metric was computed based on the differences between the predicted and observed values; more specifically, the DOY when the flight peaks were observed in the field was compared with the DOY predicted by the models. This yields four possibilities: True positives, true negatives, false positives, and false negatives, which were used to obtain the F1 metric, which is the harmonic mean of the precision and recall. The F1 was therefore calculated with the following formulas:(2)$$\mathrm{Precision}=\frac{\mathrm{TP}}{\mathrm{TP}+\mathrm{FP}},$$
(3)$$\mathrm{Recall}=\frac{\mathrm{TP}}{\mathrm{TP}+\mathrm{FN}},$$
(4)$$F1=2\times \frac{\mathrm{Precision}\;\times \;\mathrm{Recall}}{\mathrm{Precision}+\mathrm{Recall}},$$

The sites selected to validate both models from all available monitoring sites are presented in Appendix A. These sites used for the validations were randomly selected, seeking to obtain a representative sample free of bias in the choice. Validation was carried out by comparing predicted and observed values. The data corresponding to the study period from the monitoring sites listed in Appendix A was used for validation of the Touzeau model. In the same way, the monitoring sites appearing in Appendix A were used for validation of the best-performing ANN model of 4 layers of neurons.

## 5. Conclusions

The Touzeau model for *L. botrana*, based only on the temperature, showed a poor prediction performance in the area of study (three PDOs in Aragón, Northeastern Spain), with an F1 score of 0.03. In comparison, artificial neural network-type ML models, trained using data from 172 monitoring sites in the study area and considering several meteorological parameters (viz. temperature, relative humidity, wind speed, wind direction, solar radiation, and accumulated precipitation), reached an F1 score of 0.63. The best-performing model, a four-layer ANN, considered the following variables as most relevant: The Touzeau index, the Chilling index, rainfall (calculated using daily data or accumulating the half-hourly samples considering the starting date of dormancy as the start date to accumulation), and the longitude. The improvement of classical predictive models by using the reported ML models can complement pest-monitoring methods in integrated pest management, making decision-support systems more accurate.

## Figures and Tables

**Figure 1 plants-12-00633-f001:**
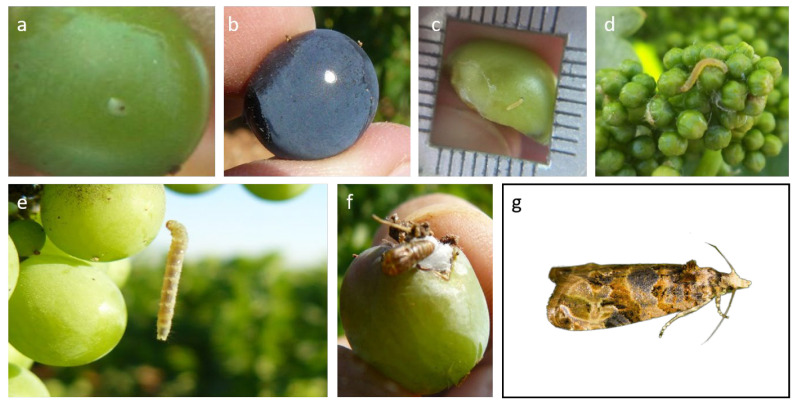
Different stages of development of *Lobesia botrana*: (**a**,**b**) Eggs; (**c**–**e**) larvae; (**f**) pupae; (**g**) adults. Photo credit: Julio Prieto-Díaz.

**Figure 2 plants-12-00633-f002:**
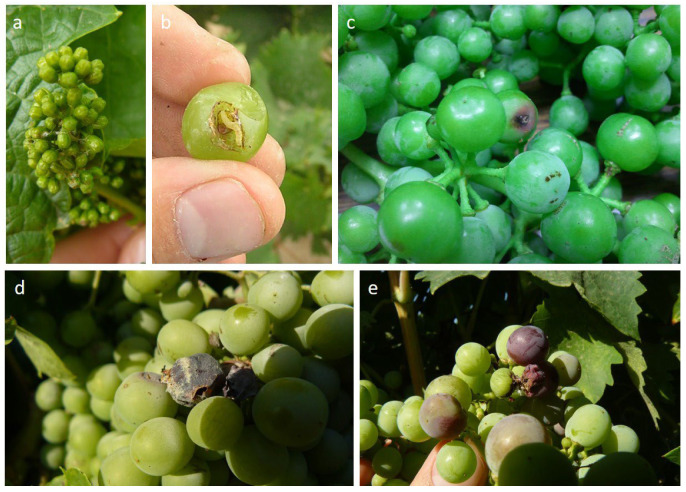
Types of damage caused by *Lobesia botrana* and associated plant diseases: (**a**) Inflorescence; (**b**,**c**) berries; (**d**,**e**) *Botrytis cinerea*. Photo credit: Julio Prieto-Díaz.

**Figure 3 plants-12-00633-f003:**
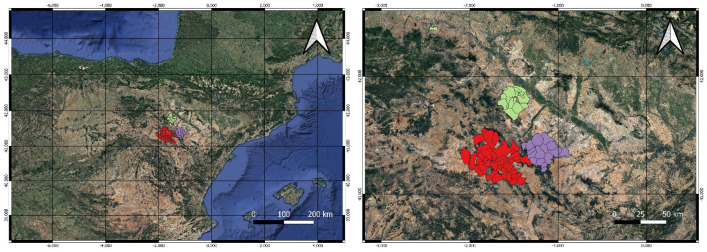
Map of the study area with the three protected designations of origin (Cariñena, in violet color; Campo de Borja, in green color; and Calatayud, in red color), in the province of Zaragoza, region of Aragón, Northeastern Spain. Map generated with QGIS [64]. Source: Government of Aragón-Geographical Institute of Aragón (https://idearagon.aragon.es/portal/, accessed 21 December 2022).

**Figure 4 plants-12-00633-f004:**
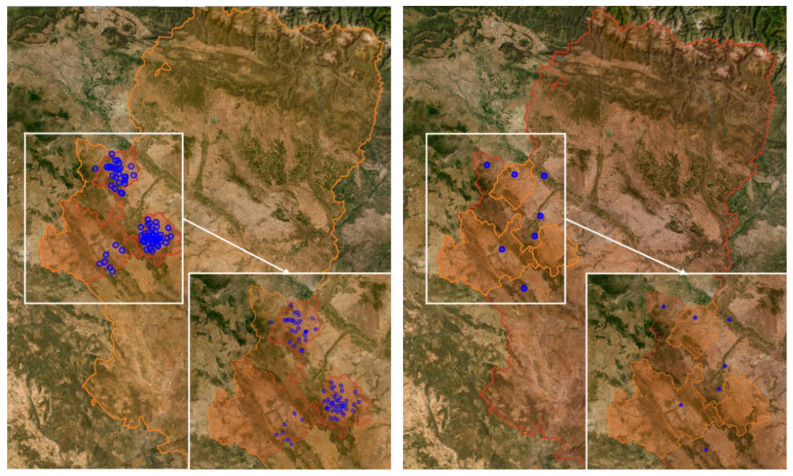
(**Left**) Map with the location of the monitoring sites; (**right**) map with the locations of the weather stations. Source: Government of Aragón-Geographical Institute of Aragón (https://idearagon.aragon.es/portal/, accessed 21 December 2022).

**Figure 5 plants-12-00633-f005:**
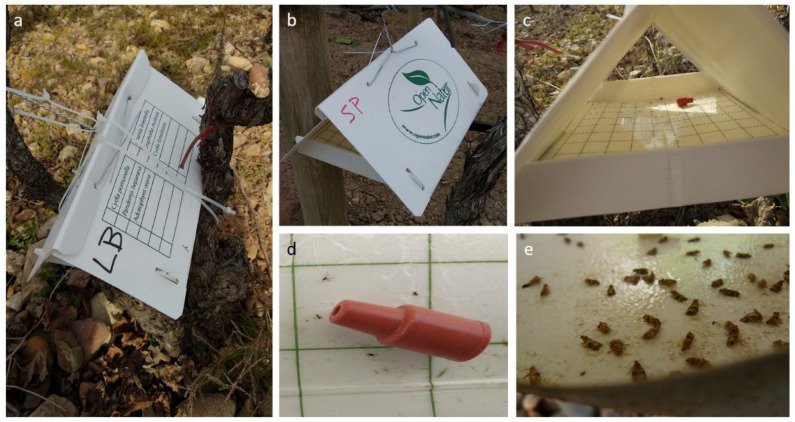
Delta traps placed in the field for flight monitoring: (**a**,**b**) Traps; (**c**) gummed bottom; (**d**) sex pheromone; (**e**) adults captured.

**Table 1 plants-12-00633-t001:** Description of the best-performing artificial neural network (ANN).

Layer	Neurons	Activation Function
0	96	linear
1	336	selu
2	64	linear
3	144	selu

**Table 2 plants-12-00633-t002:** Comparison of observed (Obs.) flight peak days of the year (DOY) with the DOY predicted by the Touzeau (Tou.) and machine learning (ML) models for the 2008−2011 seasons.

Season	Generation	Observation/ Prediction	Site Code																	
			50024A00300013	50073A03500094	50004A03400175	50073A00700043	50073A01000009	50073A02700014	50073A03400128	50073A04100021	50073A04800016	50073A05800042	50073A08400031	50073A08900011	50073A09500073	50073A10000051	50098A03000027	50201A00200124	50201A02200012	50268A00100050
2008	1st	Obs.	126	126		153					153				126			153		
		Tou.	111	111		111		111			111				111			111		
		ML	126		149	93		149			153				126			153		
	2nd	Obs.	184	184	157	159		157	157	157	159	157	157	160	159	157		159		157
		Tou.	172	172	172	172		172	172	172	172	172	172	172	172	172		172		172
		ML	186		157	157		157	157	157	159	157	157		159	157		159		156
	3rd	Obs.	221	221	224	216		224	224	224	221	224	224	224	214			216		224
		Tou.	210	210	210	210		210	210	210	210	210	210	210	210	210		210		210
		ML	220		223	215		223	223	223	221	223	223		214	223		216		223
2009	1st	Obs.	134	125	131	128		131	131	131	125	131	131	131	128			128		131
		Tou.	127	127	127	127		127	127	127	127	127	127	127	127			127		127
		ML			130	128		130	130	130	124	130	130	130	128			128		
	2nd	Obs.	175	173	152	158		152	152	152	159	152	152	152	152	152		159		152
		Tou.	166	166	166	166		166	166	166	166	166	166	166	166	166		166		166
		ML			151	157		151	151	151	159	151	151	151	152			151		
	3rd	Obs.	223	212	217	215		215	217	217	215	217	215	220	215	218		223		224
		Tou.	201	201	201	201		201	201	201	201	201	201	201	201	201		201		201
		ML			217	215		215	217	217	214	217	215	220	215			223		
2010	1st	Obs.	139	139	152	122		157	157	124		131	124	152	127	124		156	127	124
		Tou.	116	116	116	116		116	116	116		116	116	116	116	116		116	116	116
		ML	123	123	152	122		120	120	123	120	123	123	152	123	123		156	123	123
	2nd	Obs.	181	181	158	163		158	158	183	157	158	183	158	163	183		163	160	158
		Tou.	173	173	173	173		173	173	173	173	173	173	173	173	173		173	173	173
		ML	159	175	157	162		157	157	175	156	157	175	157	162	175		162	159	157
	3rd	Obs.	224	221	218	224		218	224	224	213	214	214	214	224			224	224	224
		Tou.	207	207	207	207		207	207	207	207	207	207	207	207			207	207	207
		ML	223	223	218	223		217	223	223	212	214	214	214	223			223	223	223
2011	1st	Obs.	123	123	131	128	129	131	128	128	128	128	128	128	128	128	129	121	128	128
		Tou.	103	103	103	103	103	103	103	103	103	103	103	103	103	103	103	103	103	103
		ML	123	123		126	126	130	126	126	126	126	126	126	126	126	126	121	126	126
	2nd	Obs.	164	164	157	184	157	158	158	158		182	157	182		157	157			182
		Tou.	162	162	162	162	162	162	162	162		162	162	162		162	162			162
		ML		152				157	157	157		176	150	175		150	150		150	176
	3rd	Obs.	216	216	224	220	223	214	220	220	220	224	220	220	223	213	223		223	213
		Tou.	202	202	202	202	202	202	202	202	202	202	202	202	202	202	202		202	202
		ML	216	216		216	222	210	220	220	220	223	220	220		213	222		222	213

Information for each of the site codes is available in Appendix A.

**Table 3 plants-12-00633-t003:** Hyperparameters used for training the ML model.

Hyperparameter	Value
Number of layers	3−10
Number of neurons in intermediate layers	0, 32, 512
Number of neurons in last 2 layers	16, 464
Activation functions	“selu”,”linear”,”tanh”,”softmax”
Exist activation function	sigmoid
Learning rates	10.0 × 10^−3^, 10.0 × 10^−2^, 10.0 × 10^−1^
Optimizers	‘sgd’,’adam’,’rmsprop’
Callback	Val_loss, patient = 17.0, min_delta = 0.17
Epochs	1000

## Data Availability

The data presented in this study are available upon request from the corresponding author. The data are not publicly available due to their relevance to an ongoing Ph.D. thesis.

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
