# Peer review of "Big Data and Machine Learning to Improve European Grapevine Moth (*Lobesia botrana*) Predictions"

_plants, 2023, doi:10.3390/plants12030633_

Round 1

Reviewer 1 Report

Dear Authors,

please reorder the main chapters of the paper according to the IMRAD rule, re-submit the article. In this form, the work can not be proceeded. 

Author Response

Dear Authors,

Q1. please reorder the main chapters of the paper according to the IMRAD rule, resubmit the article. In this form, the work can not be proceeded. 

Response: Thank you for your suggestion. Nonetheless, please kindly note that the order of the manuscript sections (Introduction, Results, Discussion, Materials and Methods, Conclusions) is defined by the journal’s guidelines, available at https://www.mdpi.com/journal/plants/instructions (please, see the ‘Research Manuscript Sections’ section). Consequently, we cannot rearrange them.

Reviewer 2 Report

This is a scientifically sound and original manuscript, adopting correct methodology and adequate writing.

However, necessary revisions in some sections should be made by the authors to reach a level acceptable for publication:

1. Introduction

The grapevine moth, L. botrana develops two or three annual generations depending on climatic conditions.

Line 52 refer “The grapevine moth, is one of the main pests of grapevines [9-12], responsible for high productivity losses in vineyards worldwide”.

Maybe add “in some”  - “The grapevine moth, is one of the main pests of grapevines [9-12], responsible for high productivity losses, in some vineyards worldwide. In part of the winegrowing regions it is a pest with little economic significance.

Line 56 e 57 referL. botrana has a life of several cycles; in Mediterranean latitudes, it has three generations, although there is an increasing trend towards a fourth generation [15]”.

I do not believe this. It doesn't seem possible to me to develop more than two generations in the berries. The increase in temperature also causes anticipation of maturation and there is no time to develop three generations in the grapes. You should remove that phrase, as it is nothing more than a prediction by the authors. In regions where L. botrana develops two generations, it may become three.

Article structure

Line 142 - 2. Results  

Line 162 - 3. Discussion

Line 297 - 4. Materials and Methods

In my opinion it should be reviewed and the material and methods should be presented before the results and discussion.

The article addresses an important pest in some vineyards worldwide, it was developed in three regions (Cariñena, Campo de Borja, and Calatayud), using data from several years, so we consider that after a small review it can be published.

Author Response

This is a scientifically sound and original manuscript, adopting correct methodology and adequate writing.

Response: We thank the reviewer for his/her comments and for his/her positive appreciation of the manuscript. 

However, necessary revisions in some sections should be made by the authors to reach a level acceptable for publication:

  1. Introduction

Q1. The grapevine moth, L. botrana develops two or three annual generations depending on climatic conditions.

Line 52 refer “The grapevine moth,  is one of the main pests of grapevines [9-12], responsible for high productivity losses in vineyards worldwide”.

Maybe add “in some”  - “The grapevine moth,  is one of the main pests of grapevines [9-12], responsible for high productivity losses, in some vineyards worldwide. In part of the winegrowing regions it is a pest with little economic significance.

Response: We have modified the sentence according to the Reviewer’s suggestion (in the abstract and in the third paragraph of the introduction).

Q2. Line 56 e 57 refer “L. botrana has a life of several cycles; in Mediterranean latitudes, it has three generations, although there is an increasing trend towards a fourth generation [15]”.

I do not believe this. It doesn't seem possible to me to develop more than two generations in the berries. The increase in temperature also causes anticipation of maturation and there is no time to develop three generations in the grapes. You should remove that phrase, as it is nothing more than a prediction by the authors. In regions where L. botrana develops two generations, it may become three.

Response: We mentioned this based on previous works such as [15]. Anyway, as this is not really important for our paper, we have removed the sentence, according to the Reviewer’s suggestion.

Article structure

Q3. Line 142 - 2. Results  

Line 162 - 3. Discussion

Line 297 - 4. Materials and Methods

In my opinion it should be reviewed and the material and methods should be presented before the results and discussion.

Response: Thank you for your suggestion. Nonetheless, please kindly note that the order of the manuscript sections (Introduction, Results, Discussion, Materials and Methods, Conclusions) is defined by the journal’s guidelines, available at https://www.mdpi.com/journal/plants/instructions (see ‘Research Manuscript Sections’ section). Consequently, we cannot rearrange them.

The article addresses an important pest in some vineyards worldwide, it was developed in three regions (Cariñena, Campo de Borja, and Calatayud), using data from several years, so we consider that after a small review it can be published.

Response: We thank the Reviewer for his/her positive feedback on the study.

Reviewer 3 Report

This is a well-structured paper that will have an impact on the actual pest management strategy on the vineyards aginst L. botrana. I have some minor recommendations on the writing style. Some acronyms are mention but described on later pages. It is recommended to improve quality of the photos of figure 1. Also, I have some questions included in the attached file. 

Author Response

This is a well-structured paper that will have an impact on the actual pest management strategy on the vineyards aginst L. botrana.

I have some minor recommendations on the writing style.

Response: We thank the reviewer for his/her comments and for his/her positive appreciation of the manuscript. 

Q1. Some acronyms are mention but described on later pages.

Response: We thank the Reviewer for pointing out this issue. We have revised the paper and all abbreviations are now defined upon first use.

Q2. It is recommended to improve quality of the photos of figure 1.

Response: We thank the Reviewer for his/her suggestion. We have replaced image (g) with one of better quality and improved the resolution of the figure.

Also, I have some questions included in the attached file. 

Review plants-2147490. Big Data and Machine Learning to Improve European Grape vine Moth (Lobesia botrana) Predictions. Line Observations

Q3. 60-61 Figure 1, could show more magnifications of images of each biological moth stages to visualize morphological details

Response: We thank the reviewer for his/her suggestion. We have improved the figure, according to the Reviewer’s suggestion.

Q4. 70-72 After botrana, add: and associated plant disease:

Response: We have added the text indicated by the Reviewer.

Q5. 79 What types of control measures are programmed against L. botrana?

Response: In the region of Aragon, the pest L. botrana is controlled almost exclusively using mating disruption. If necessary, a phytosanitary treatment is carried out on an ad hoc basis. This has been noted later on in the introduction, after discussing sex pheromone-based mating disruption techniques (in our view, including this explanation in L79 would disrupt the discursive line of the text). We have added: “(like the sex pheromone-based mating disruption technique, mentioned later in this section, or phytosanitary treatments, when necessary)”.

Q6. 123 Add: pest, before: population

Response: We have added ‘pest’ before ‘population’, as suggested by the Reviewer.

Q7. 131-132 Any idea of what factors may cause the sex pheromone-based mating disruption system became ineffective? Because in lines 114-115 you mention this technique has been successfully implemented for L. botrana control.

Response: At present, the mating disruption method is still effective, but in other pests it has been observed that males learn to differentiate between the pheromone of the diffusers and that of the females, which is much more complex. A new sentence has been included in the text, making it clear that in L. botrana this effect has not occurred yet. We have added: “(at present, the mating disruption method is still effective, but in other pests it has been observed that males learn to differentiate between the pheromone of the diffusers and that of the females, which is much more complex)”.

Q8. 158 Meaning of DOY?

Response: DOY stands for ‘Day of the year’. In the revised version of the paper, we have made sure that all abbreviations are now defined upon their first appearance.

Q9. 246 After protection measures, add: against L. botrana…

Response: We have added it.

Q10. 251 After botrana, add: population density

Response: We have added it.

Q11. 254 Again, see observations in lines 131-132.

Response: Please kindly refer to the response to Q7. Another clarification has been included in the text in the indicated line, which now reads: “[…] where the mating disruption method using sex pheromones has been implemented (which, as noted above, is still effective), […]”

Q12. 272 After: economic, add: and ecological…

Response: We have added it.

Q13. 275 After: predicting, add: pest or

Response: We have added it.

Q14. 284 You are referring only to the right time to initiate insecticide application to control a pest using AI-based models, but what other tactics in an IPM program could be used to ensure a lower environmental impact?

Response: Sexual confusion, which is the most effective method and the one that has the lowest environmental impact. A clarification has been included at the end of the paragraph: “Despite the outlined room for improvement of AI-based models, the process and results presented in this work highlight the benefits of ML applied to plant health strategies and its potential contribution to increasing the sustainability of agricultural activity, through a lower environmental impact as well as a lower economic cost of production (e.g., as a result of determining the right time to initiate insecticide application to control a pest or, preferably, of using IPM tactics such as sexual confusion)”.

Q15. 291-292 Testing or validate how the ML models perform with more current data, directly in the vineyards for how many seasons?

Response: The sentence has been rewritten, providing further details. It now reads: “[...] one will be validating how the ML models perform with data collected after the implementation of the mating disruption method (i.e., from 2012 to 2022).

Q16. 328-329 Sources of the figure 4.

Response: We thank the reviewer for pointing this out. In the revised version of the paper, we now mention the source of the figure (Government of Aragón - Geographical Institute of Aragón, https://idearagon.aragon.es/portal/).

Q17. 337 Name of the supplier?

Response: In the revised version of the paper, the name of the supplier has been added “[…] (OpenNatur SL, Lleida, Spain)”.

Q18. 393 Add: seasonal, before: behavior

Response: We have added it.

Q19. 399 Replace: this, with: the present

Response: We have replaced it.

Q20. 663-665 Year of the publication? It has a link: https://www.scitepress.org/Papers/2020/101329/101329.pdf, and DOI: 10.5220/0010132900750082

Response: The requested information has been added, together with the DOI. We have also revised the rest of the references in the paper.

Q21. 664 It is repeated: Proceedings of

Response: In the revised version of the paper, we have corrected this redundancy.

Reviewer 4 Report

An interesting article on developing a predicative model for grape pest. The science is sound. The main issue, which is minor, is that there is a great deal of introductory material in the discussion section. To avoid making the introduction section too lengthy when moving this information into it, I suggest reducing the introductory material on the pest insect to one or two paragraphs. As written, it is quite a comprehensive review, which is unnecessary in a research article. I also suggest eliminating the subheadings in the discussion and writing it in a continuous flow. 

Author Response

An interesting article on developing a predicative model for grape pest. The science is sound.

Response: We thank the reviewer for his/her comments and for his/her positive appreciation of the manuscript. 

Q1. The main issue, which is minor, is that there is a great deal of introductory material in the discussion section. To avoid making the introduction section too lengthy when moving this information into it, I suggest reducing the introductory material on the pest insect to one or two paragraphs. As written, it is quite a comprehensive review, which is unnecessary in a research article.

Response: We have simplified the discussion section. For this purpose, we have removed the first paragraph of subsection 3.2, the entire subsection 3.3, and lines 269-282 from subsection 3.5, as well as combined the last two subsections into one. We have also reduced the introductory material on the pest insect by removing text from lines 57-59. We believe that the amount of relevant introductory material is now more balanced.

Q2. I also suggest eliminating the subheadings in the discussion and writing it in a continuous flow. 

Response: Given that the discussion is lengthy and covers different aspects that are not directly linked to each other, we would rather keep the subsections’ headings. In our view, the use of these subheadings helps to categorize your interpretations into themes. Nonetheless, taking into account the comment provided by the Reviewer, we have simplified the structure by removing a subsection (3.3) and combining the last two subsections into one.

Q3. Line 56: L. botrana has a life of several cycles à L. botrana is multivoltine

Response: We have corrected this, as suggested.

Q4. Line 79: and programming control à and applying control measures

Response: The text has been reworded according to the Reviewer’s suggestion.

Q5. Line 84: between controls à between treatment applications

Response: ‘Controls’ has been replaced with ‘field inspections’ to preserve the intended meaning.

Q6. Line 141 (Results): This section is extremely short and needs to be expanded. Including actual peak dates compared to what the models predicted would be interesting, for example.

Response: We have added Table 2, which shows the comparison of observed flight peak DOY with the DOY predicted by the Touzeau and ML models for the 2008 to 2011 season.

Q7. Lines 183-189: This is introductory material and belongs in the introduction section. It can be summarized here, but should not be introduced for the first time in the discussion section.

Response: Taking into account the suggestion provided by the Reviewer, we have removed this first paragraph of subsection 3.2.

Q8. Lines 214-215: encountered that à either found that or determined that

Response: We have corrected this, as suggested.

Q9. Lines 233-242: This is introductory material and should be moved to the introduction. In the discussion section, discuss how the architecture of your ANN model compares to the ideal.

Response: Taking into account the suggestion provided by the Reviewer, we have removed subsection 3.3.

Q10. Line 248: pest affection. What is a pest affection? I think this is a mistranslation.

Response: We thank the Reviewer for pointing this out. In the revised version of the paper, ‘affection’ has been replaced by ‘infestation’.

Q11. Line 248: thinking on the field implementation of measures. How would thinking about field implementation of measures accomplish anything? Should a model indication of a developing pest problem prompt action, not just thinking about action?

Response: This may be a misunderstanding of what was intended to be expressed here, probably due to a mistranslation. We believe that changing “thinking on…” to “in terms of…” will clarify the sentence. In the revised version of the paper, it now reads: “…in terms of field implementation of measures…”.

Q12. Lines 269-282: Again, a lot of this is introductory material.

Response: We have removed these lines, which are part of subsection 3.5.

Q13. Line 319: on this pest à on pest

Response: The word ‘this’ has been deleted, as suggested.

Q14. Line 346: captures tends to zero à captures trends to zero.

Response: ‘Tends’ has been replaced with ‘trends’, as suggested.

Q15. Line 346: captures à capture

Response: This has been corrected in the revised version of the paper, as suggested.

Q16. Line 489: below. Where below?

Response: We apologize for the oversight. A reference to Tables S4 and S5 has been included.

Round 2

Reviewer 1 Report

I accept your new version